# Structural basis of epitope selectivity and potent protection from malaria by PfCSP antibody L9

Gregory M. Martin[1], Monica L. Fernández-Quintero [2], Wen-Hsin Lee [1], Tossapol Pholcharee [1,6], Lisa Eshun-Wilson[1], Klaus R. Liedl[2], Marie Pancera [3], Robert A. Seder [4], Ian A. Wilson [1,5] & Andrew B. Ward [1] ✉

A primary objective in malaria vaccine design is the generation of high-quality antibody responses against the circumsporozoite protein of the malaria parasite, *Plasmodium falciparum* (PfCSP). To enable rational antigen design, we solved a cryo-EM structure of the highly potent anti-PfCSP antibody L9 in complex with recombinant PfCSP. We found that L9 Fab binds multivalently to the minor (NPNV) repeat domain, which is stabilized by a unique set of affinity-matured homotypic, antibody-antibody contacts. Molecular dynamics simulations revealed a critical role of the L9 light chain in integrity of the homotypic interface, which likely impacts PfCSP affinity and protective efficacy. These findings reveal the molecular mechanism of the unique NPNV selectivity of L9 and emphasize the importance of anti-homotypic affinity maturation in protective immunity against *P. falciparum*.

Malaria remains one of the world's deadliest infectious diseases, and in 2021 was responsible for 241 million clinical infections and 627,000 deaths worldwide (WHO, 2021), primarily among young children in sub-Saharan Africa. RTS,S/AS01B (RTS,S), the only approved malaria vaccine, is only partially effective, providing ~30% protection from clinical infection after four years in children aged 5–17 months[1,2]. Thus new tools, like next-generation vaccines and highly potent monoclonal antibodies (mAbs), the latter of which can provide prolonged, sterilizing immunity[3–5], are needed for prevention and elimination of malaria.

PfCSP, the primary surface antigen of *P. falciparum* sporozoites, is a major target for vaccines and mAbs as it is both highly conserved and critical for the initiation of malaria infection. PfCSP contains an immunodominant central repeat region composed of repeating four amino-acid units, structurally defined as DPNA, NPNV, and NPNA[6–13]. These roughly define the junctional, minor repeat, and major repeat epitopes, respectively. Each epitope can generate potent antibodies

that prevent malaria infection in animal models[14–16], with the junctional mAb CIS43LS demonstrating high-level protection against controlled human malaria infection (CHMI) in humans[3,5]. Recently, we identified the minor repeat-specific mAb L9 as one of the most potent anti-PfCSP mAbs isolated to date[17], which can also confer high-level sterilizing immunity against CHMI in humans[4]. Like many of the most potent NPNA-specific mAbs, L9 is encoded by the *IGHV3-33/IGKV1-5* heavy/light chain gene combination. However, L9 is highly specific for the NPNV (minor) repeats and relies on critical contributions from the light chain for both NPNV selectivity and high potency[8].

Here we used cryo-EM to understand the molecular basis of these unique functional properties. We demonstrate that L9 utilizes a distinct homotypic interface to stabilize multivalent Fab binding to the PfCSP minor repeats, and a unique paratope structure to selectively interact with NPNV repeats. In combination with MD simulations, these data indicate a key role in affinity-matured homotypic contacts in the

[1]Department of Integrative Structural and Computational Biology, The Scripps Research Institute, La Jolla, CA 92037, USA. [2]Department of General, Inorganic, and Theoretical Chemistry, Center for Chemistry and Biomedicine, The University of Innsbruck; Innrain 80-82/III, 6020 Innsbruck, Austria. [3]Vaccine and Infectious Disease Division, Fred Hutchinson Cancer Research Center, Seattle, WA 98109, USA. [4]Vaccine Research Center, National Institute of Allergy and Infectious Diseases, National Institutes of Health, Bethesda, MD 20892, USA. [5]The Skaggs Institute for Chemical Biology, The Scripps Research Institute, La Jolla, CA 92037, USA. [6]Present address: Department of Biochemistry, University of Oxford, Oxford OX1 3DR, UK. ✉e-mail: andrew@scripps.edu

L9 light chain for mediating high-affinity PfCSP binding and potent protection from malaria infection.

## Results

### L9 binds multivalently to the PfCSP minor repeats

For structure solution, a recombinant PfCSP construct was used, rsCSP, that contains the full N-terminal, junctional, minor repeat, and C-terminal regions, and about half the number of NPNA repeats as the 3D7 reference strain (Fig. 1a). To overcome both aggregation and preferred orientation of the L9 Fab-rsCSP complex in vitreous ice (see Methods), a large cryo-EM dataset was collected which resulted in a 3.36 Å resolution reconstruction (Fig. 1b; Supplementary Fig. 1; Supplementary Table 1). In the cryo-EM map, we observe three tightly packed Fabs bound to a central rsCSP, with each Fab simultaneously interacting with the peptide and the adjacent Fab via homotypic interactions[10,13,18,19]. In general, the complex is homogeneous and the density is well-resolved for each L9 variable region (Fv) as well as the rsCSP peptide (Fig. 1b). The structure of rsCSP, built de novo based on the EM density, consists solely of the minor repeat region (Fig. 1f). The modeled antigen sequence comprises 26 residues encompassing three complete NPNV and DPNA repeats, i.e., NA(NPNVDPNA)$_3$; there is no additional density observed that would correspond to N-terminal, C-terminal, or major repeat regions. Moreover, we did not identify any 2D or 3D classes with more than three Fabs, indicating that any potential binding of L9 to the NPNA repeats was not stable enough to be captured by cryo-EM (Supplementary Fig. 2b); this is further supported by biolayer interferometry data showing rapid dissociation of L9 Fab to an NPNA-only peptide (NPNA$_8$; Supplementary Fig. 2c, d). The L9 Fab and peptide cryo-EM structures correspond well with our recent X-ray structures of two chimeric precursors of L9 (L9$_K$/F10$_H$ and F10$_K$/L9$_H$) in complex with a short minor repeat peptide (NA<u>NPNVDP</u>)[8] (Supplementary Fig. 3a–d). Relative to a representative Fv (Fab B) in the L9 cryo-EM structure, Cα RMSD values for both chimeric Fvs are ~0.5 Å, and ~0.1 Å when comparing only the PfCSP peptide encompassing the NPNV repeat. Within the L9-rsCSP complex, there is also a high degree of similarity between repeating components, with Cα RMSD values of ~0.5 Å between the three L9 Fvs, and 0.05–0.10 Å between the three NPNV epitopes on rsCSP (Supplementary Fig. 3e–g).

### L9 utilizes a distinct paratope structure to confer NPNV selectivity

In the cryo-EM structure, each L9 Fab primarily engages a single NPNV repeat, while the DPNA repeats are largely unbound and serve as a linker between each NPNV (Fig. 1e, g). This binding site model is strongly supported by our previous work demonstrating the high selectivity of L9 for NPNV over both DPNA and NPNA repeats[8,17], and by the structural data itself, as alternate registrations of rsCSP produced substantially worse fits to the cryo-EM map (Supplementary Fig. 4). Thus, the full epitope bound by a single L9 Fab is NPNVD (Fig. 1g). Each NPNV motif adopts a type-1 β-turn, which is frequently observed for DPNA and NPNA motifs bound to anti-PfCSP antibodies from a variety of heavy and light chain lineages[6,10,20]. The DPNA repeats in the L9 structure, however, are more extended and lack clear secondary structure elements (Fig. 1f, h). The L9 epitope is centered on the NPNV type-1 β-turn, which resides in a deep, central pocket on the Fab formed primarily from CDRL1, CDRL3, and CDRH3, with smaller contributions from CDRH1 and H2 (Fig. 2a; Fig. 3a, b). Interestingly, overall buried surface area (BSA) on L9 is slightly biased toward the light chain (LC; L9$_K$) (Fig. 3a, b). Of the 550 Å$^2$ total BSA on a single L9 Fab, L9$_K$ contributes 294 Å$^2$ (53.5%), while the heavy chain (HC; L9$_H$) contributes 256 Å$^2$ (46.5%), indicating a critical role of L9$_K$ in PfCSP binding.

As frequently observed in anti-NPNA major repeat mAbs, many direct antigen contacts are with germline-encoded aromatic residues, which in L9 create a hydrophobic cage surrounding the NPNV motif (Fig. 2b, Supplementary Table 2). In particular, W32$^L$ in CDRL1 stacks

closely against the N-terminal Asn of the NPNV motif (N1) forming a CH-π bond, while Y94$^L$ in CDRL3 engages the repeat Pro (P2) (Fig. 2c, Fig. 3c, d). L9 also utilizes the strictly conserved *IGHV3-33* germline residue W52$^H$ in CDRH2, which in all structures of *IGHV3-33* mAbs solved to date forms a critical CH-π interaction with P6 of the second NPNA repeat in the NPNA$_2$ epitope[9,10,13,20]. However, in L9, this role is assumed by Y94$^L$, and W52$^H$ principally acts to stabilize the Y94$^L$:P2 interaction through a π-π stacking interaction with the Y94$^L$ side chain (Fig. 3d–f).

This paratope structure is distinct from most other *IGHV3-33* mAbs targeting both major and minor repeats. In L9, a repositioning of the HC and LC CDR3 loops, along with a rearrangement of W52$^H$ and CDRH2, creates a compact, central PfCSP binding pocket bounded by each of the HC and LC CDRs (Fig. 2b, e). A somatically mutated residue, R96$^L$ in CDRL3, is found at the base of the pocket and creates a highly basic cavity (Fig. 3g) that is nearly fully occupied by the N3 side chain, which forms key H-bonds with R96$^L$ (Fig. 2c). V4 occupies a hydrophobic cavity at the interface of CDRL1, L3, and CDRH3 (Fig. 3c), forming hydrophobic contacts with the side chain of Y97$^H$. Notably, Y97$^H$ accounts for the greatest amount of BSA in the L9 paratope of any HC residue (Fig. 3a), and this largely stems from the interaction with V4, suggesting that this CDRH3 residue is critical for selectivity of L9 for NPNV over NPNA. Comparison of the paratope structure of L9 to a panel of NPNA-targeting *IGHV3-33/IGKV1-5* mAb structures suggests that, in addition, the unique arrangement of the L9 CDR loops is unfavorable to NPNA binding in this conformation, as superimposition of these Fab-NPNA$_2$ cryo-EM and X-ray structures onto the L9 Fab structure revealed extensive clashing between the peptide and the L9 CDRH1, CDRH3, and CDRL3 loops (Supplementary Fig. 5b, c).

### Unique homotypic interactions stabilize multivalent PfCSP binding

Another unique property of L9 is the ability to "crosslink" two NPNV motifs within the minor repeat region of PfCSP, which improves binding affinity[8]. Our cryo-EM structure reveals that L9 achieves this through multivalent Fab binding to sequential NPNV repeats stabilized by an extensive antibody-antibody, or homotypic, interface between adjacent Fabs (Fig. 4a). Homotypic interactions have now been identified in several anti-NPNA mAbs and appear to be a characteristic feature of the *IGHV3-33* antibody family[10,13,18–20]. Importantly, we demonstrate L9 as a non-NPNA-targeting anti-PfCSP mAb to also utilize homotypic interactions, suggesting that both the major and minor PfCSP repeats can facilitate their development.

The L9 homotypic interface is distinct from that observed in NPNA-specific *IGHV3-33* mAbs, which is generally conserved and derives primarily from the heavy chain[13,19] (Supplementary Fig. 7). In contrast, L9$_K$ contributes numerous critical homotypic contacts, and total BSA in the interface is evenly distributed between heavy and light chains (905 Å$^2$ and 839 Å$^2$, respectively) (Fig. 4e, f). In the cryo-EM structure, L9$_K$ of FabC packs tightly against L9$_H$ of Fab B, and extensive polar and hydrophobic contacts are made between CDRL1 and the LC framework region 3 of FabC (LFR3) with HFR1, CDRH1, and CDRH3 of Fab B (Fig. 4b–d; Supplementary Fig. 6; Supplementary Table 3). The homotypic interface between Fab B and Fab A is nearly identical. Importantly, several residues mediating critical homotypic interactions (Fig. 4b–d) correlate with somatic hypermutation of the germline *IGHV3-33* and *IGKV1-5* genes (Fig. 4e, f; Supplementary Fig. 7). Four somatically mutated residues in L9$_K$, F28$^L$ and R31$^L$ in CDRL1, and E68$^L$ and H70$^L$ in LFR3, account for the majority of BSA contributed by the LC to the homotypic interface (Fig. 4e).

E68$^L$ lies at the core of the homotypic interface in L9, where it forms a key salt bridge with the germline-encoded R94$^H$ of CDRH3$_B$ (Fig. 4b; Supplementary Fig. 6). In L9$_H$, R94$^H$ forms a conserved interaction with Y102$^H$ to stabilize the base of CDRH3; thus E68$^L$ may also indirectly impact antigen binding through stabilization of the CDRH3 loop in the adjacent Fab. F28$^L$ coordinates a series of π-π stacking

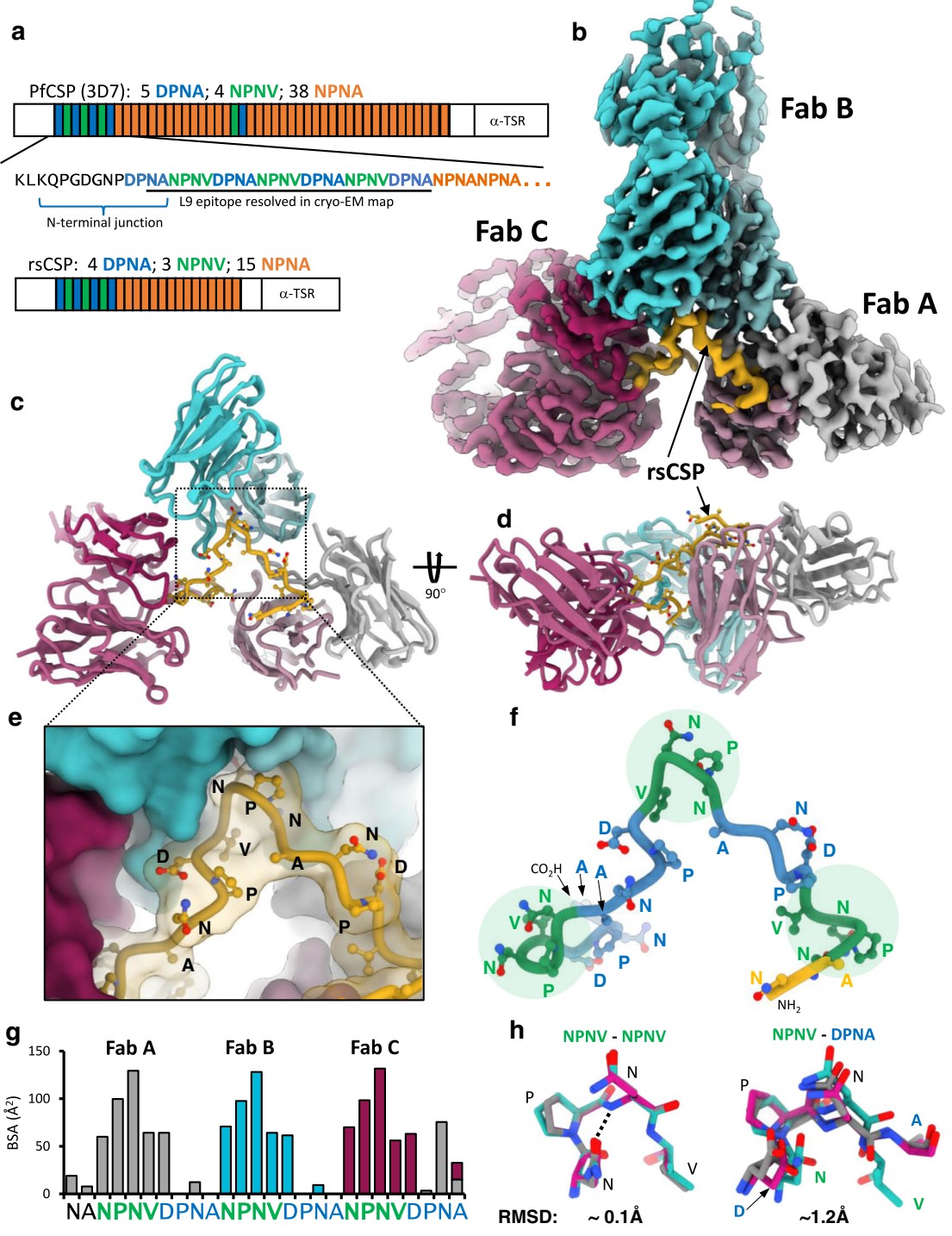

**Fig. 1 | Cryo-EM structure of the L9 Fab-rsCSP complex. a** Schematic of protein sequence of full-length PfCSP and rsCSP (recombinant). Each box corresponds to a single repeat. The minor repeat region is in blue and green. **b** Cryo-EM map of L9-rsCSP at 3.36 Å. **c** Ribbon diagram of the atomic model; only the Fab variable region (Fv) was built into the density. **d** Rotated view of **c**. **e** Zoomed-in view of **c**, shown in a surface representation. **f** Model of the minor repeat peptide, colored as in a. NPNV type-1 β-turns are highlighted with a green circle. **g** Buried surface area on rsCSP, color-coded to the Fab with which each rsCSP residue interacts. **h** Alignment of the three NPNV motifs (left), or the three DPNA motifs aligned to the central NPNV motif (right). RMSD: root mean square deviation. Source data are provided as a Source Data file.

interactions in the opposing CDRH1$_B$ (Y32$^H$) and CDRH3$_B$ (F96$^H$ and F100c$^H$) while also packing against the E68$^L$ side chain. This pi network culminates in a cation-π bond between R31$^L$ from CDRL1$_C$ and F100c$^H$ from the opposing CDRH3$_B$ (Fig. 4c). On the other side of the homotypic interface from E68$^L$, a mutated framework residue H70$^L$ forms a hydrogen bond with the side chain of Q1$^H$ in Fab A in addition to

multiple van der Waals contacts with CDRH1$_B$ (Fig. 4d). Each of these homotypic contacts is not encoded in the germline sequence, and none directly contact rsCSP (Fig. 3a, b). These findings provide strong evidence for affinity maturation to optimize antibody-antibody binding, which may, in turn, enhance PfCSP avidity and protective efficacy, as we have shown recently for multiple NPNA-specific *IGHV3-33* mAbs[13].

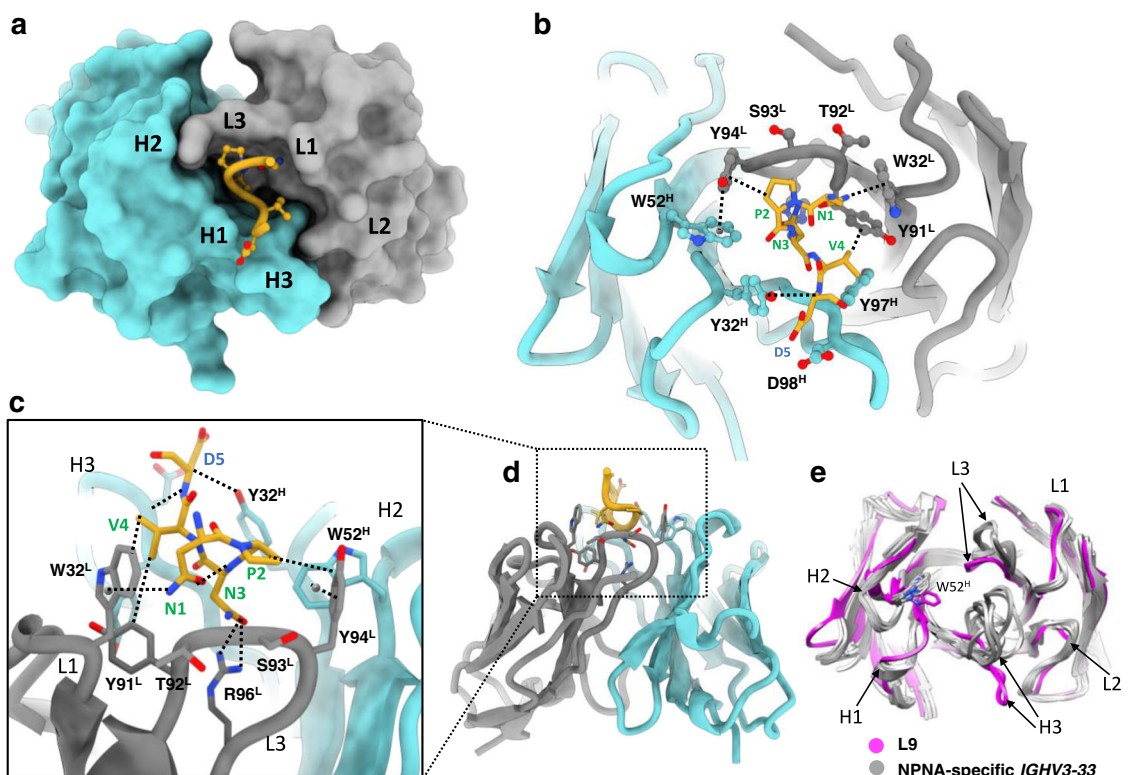

**Fig. 2 | The L9 PfCSP epitope comprises NPNVD. a** Surface representation of L9 Fab, with central NPNVD shown in gold. Heavy and light chain CDR loops are specified as H1, L1, etc. **b** Structural details of PfCSP binding pocket. Key interactions are highlighted with dashed lines. **c** Rotated view of **b**, zoomed-in from **d**. **d** Rotated view of **a**, shown in ribbon diagram. **e** Alignment of L9 Fab (magenta) with a panel of NPNA-specific *IGHV3-33* Fabs; sequences in Fig. S7. Note that in the main text and in figures, we use the nomenclature L9$_K$ and L9$_H$ to refer to the light chain and heavy chain, respectively, as a whole. When referring to specific amino acids within either chain, we use the more general notation of, for example, W32$^L$ and W52$^H$.

The four somatic mutations in L9$_K$ are atypical: F28$^K$, E68$^K$, and H70$^K$ are observed in <1% percent of all human *IGKV1* light chain sequences, while R31$^K$ is observed in only 2% (Supplementary Fig. 7a)[21]. Strikingly, F28 and H70 also correspond to two of the five amino-acid differences between mature L9$_K$ and the light chain of a clonal relative and precursor of L9, F10$_K$ (S28 and D70 in F10$_K$). Previously, we demonstrated the critical role of L9$_K$ in PfCSP binding and potency of protection by forming chimeric mAbs of L9 and F10, in which the light or heavy chain of L9 was paired with heavy or light chain of F10 (L9$_K$/F10$_H$ and F10$_K$/L9$_H$). Specifically, we found key functional differences between L9 and the F10$_K$L9$_H$ chimera: (1) reduced avidity to PfCSP minor repeats, (2) loss of the ability to bind two adjacent NPNV repeats, and (3) significantly reduced protection in vivo ($p < 0.001$)[8]. We also recently showed that mutation of residues mediating key homotypic interactions in a family of potent NPNA-specific *IGHV3-33* mAbs caused similar functional effects as see in F10$_K$ chimera relative to L9[13]. Thus, as F28 and H70 both mediate key homotypic interactions in L9, which would likely be lost in F10$_K$, these residues may be key determinants in the minor repeat specificity and exceptional potency of L9.

**Evolved homotypic contacts in L9$_K$ are critical for complex stability**

To test this hypothesis, and to understand the role of homotypic contacts in L9$_K$ in general, we used molecular dynamics simulations to characterize WT L9 and a series of L9$_K$ variants. L9$_K$ residues were reverted to either the germline *IGKV1-5* gene (R31S, E68G, H70E) or to the L9$_K$ precursor F10$_K$ (F28S, H70D). We first compared the free energy landscapes of the CDR loops of individual Fv domains unbound to rsCSP (Fig. 5; Supplementary Fig. 8). We find that the R31S, E68G, and H70D/E mutations in L9$_K$ result in a broader conformational space and additional highly probable minima compared to the WT L9 Fv,

indicating that these residues are critical for determining the shape and the conformational flexibility of the paratope (Fig. 5b, c; Supplementary Fig. 8). These minima correspond to a substantial shift away from the binding competent conformation in combination with a higher conformational entropy, suggesting a decrease in stability and/or binding affinity (Fig. 5d). Importantly, when combined (R31S-E68G-H70D), MD simulations of the trimeric structure in complex with rsCSP predict that these mutations significantly destabilize the homotypic interface (Supplementary Table 4; $p < 0.001$), indicating their key role in mediating homotypic interactions. Interestingly, in the context of the trimeric complex, the H70D single mutant is predicted to stabilize homotypic interactions (Supplementary Table 4), suggesting the germline E70 or F10$_K$ D70 may have initialized the evolution of homotypic interactions during L9 maturation. Unlike other LC mutants, the F28S Fv reveals a similar conformational space and diversity in the CDR loops compared to the WT L9 Fv. However, simulations of F28S show the formation of a new *intra*molecular salt bridge between residues R31$^L$ and E68$^L$, with simultaneous loss of the *inter*molecular salt bridge between E68$^L$ and R94$^H$ and the cation·π bond between R31$^L$ and F100c$^H$ (Fig. 5a). These results suggest that, in addition to direct homotypic interactions, F28 acts indirectly through E68$^L$ and R31$^L$ to further stabilize antibody-antibody binding. This is reflected in simulations that predict significantly decreased interaction energies of the homotypic interface in the F28S mutant relative to WT L9 (Supplementary Table 4); this is visualized in Movie S1.

To understand the molecular basis of key functional differences between L9 and F10, we next performed MD simulations of the F10 chimeras in the context of the trimeric Fab-rsCSP complex. Compared to WT L9 and L9$_K$/F10$_H$, simulations predict that the homotypic interface is strongly destabilized in the F10$_K$/L9$_H$ chimera (Supplementary Table 4). This suggests that F10$_K$/L9$_H$ would not bind

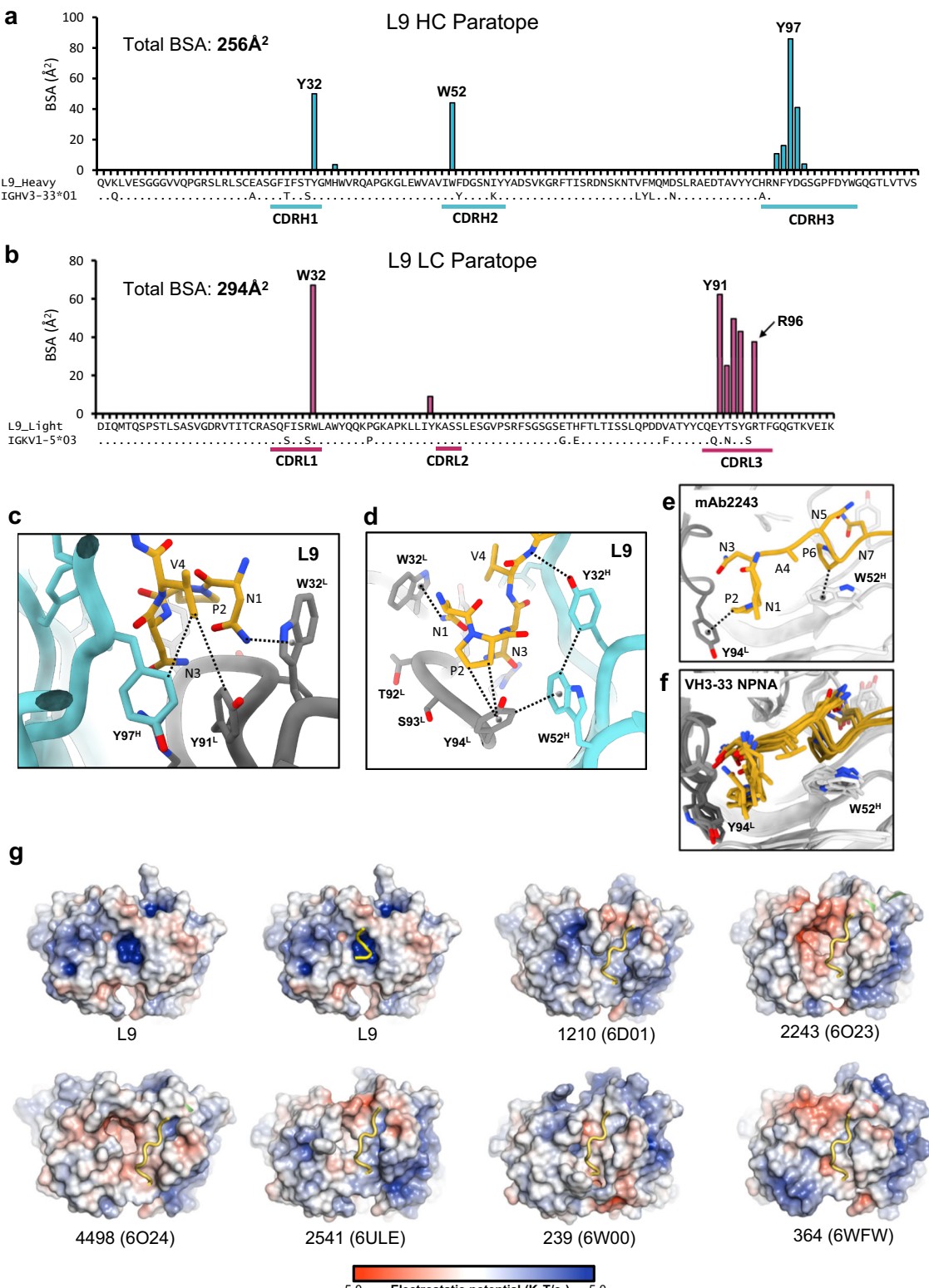

**Fig. 3 | Structural details of the L9 paratope. a** Buried surface area (BSA) contributions of individual residues to rsCSP binding in the L9 heavy chain. Sequence alignment to the *IGHV3-33* germline gene shown below. **b** Same as in a, for the L9 light chain. **c, d** Structural details of NPNV binding. **e** NPNA₂ epitope structure in the NPNA-specific mAb 2243 (PBD 6O23), highlighting the two key CH-π interactions of germline-encoded aromatic residues (W52ᴴ and Y94ᴸ) with the repeat prolines. **f** Same as in e, with X-ray structures of six NPNA-specific mAbs superimposed to

highlight structural conservation. These six mAbs are shown in **g**. **g** Electrostatic surface potentials from L9 cryo-EM structure +/− peptide (upper left two panels) and X-ray structures of six other NPNA-specific mAbs bound to peptide; electrostatic potentials were calculated in PyMol[61]. The PDB accession codes are in parentheses. $K_b$: Boltzmann constant; $T$: temperature in kelvin; $c_e$: electron charge in coulombs. Source data are provided as a Source Data file.

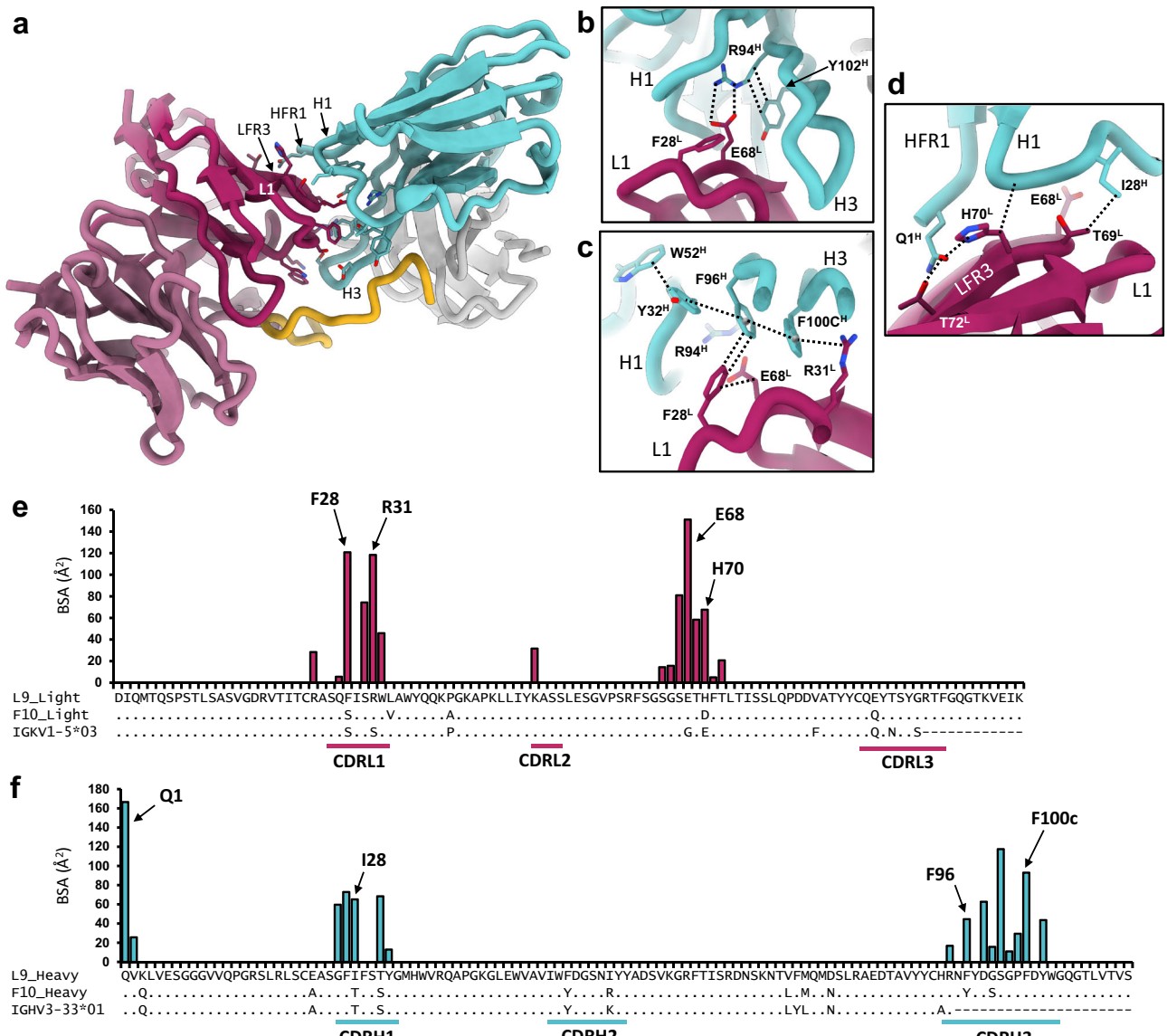

**Fig. 4 | L9$_K$ mediates extensive homotypic interactions. a** Ribbon diagram of Fab B (cyan) and C (maroon); side chains of interacting residues are shown. **b–d** Structural details of key homotypic interactions. Dashed lines indicate specific contacts. **e** Buried surface area (BSA) contributions of individual residues to the homotypic interface in L9 light chain. Sequence alignment with F10$_K$ and germline *IGKV1-5* gene is shown below. **f** Same as in **e**, for L9 heavy chain, with sequence alignment to F10$_H$ and germline *IGHV3-33* gene. Source data are provided as a Source Data file.

multivalently to the minor repeats and would have overall reduced binding affinity, which is consistent with our previous functional data on this chimera[8]. Five residues differ between L9$_K$ and F10$_K$: F28S, L33V, P40A, H70D, and E90Q (Fig. 4e). We find that the F28S mutation alone accounts for ~80% of the predicted destabilization of the homotypic interface observed with F10$_K$/L9$_H$ compared to WT L9, while the H70D single mutant and the L33V-P40A-E90Q triple mutant Fvs are predicted to both slightly increase stability of the complex. Taken together, these data suggest that the dramatic destabilization seen in MD simulations of the F10$_K$/L9$_H$ chimera is primarily the result of the F28S mutation. Therefore, this rare mutation in L9$_K$ (S28F), and the network of homotypic contacts it mediates, may underlie the key functional differences between L9 and F10$_K$/L9$_H$.

## Discussion

This study reveals the structural basis for the extraordinary selectivity and binding affinity of L9 for the NPNV minor repeats and highlights the critical role of L9$_K$ for both functions. We find that rare, somatically

mutated residues in L9$_K$ mediate extensive homotypic contacts between adjacent L9 Fabs and thus multivalent binding to adjacent NPNV motifs. These contacts underscore the requirement of at least two NPNV motifs for high affinity PfCSP binding by L9 (1000 nM vs 13 nM for peptides with one and two NPNV, respectively)[8]. Based on our recent finding that affinity-matured homotypic interactions in three potent NPNA-specific *IGHV3-33* mAbs are critical for both high NPNA avidity and protective efficacy[13], it is likely that L9$_K$-mediated homotypic interactions are also critical for the potency of L9. Notably, these L9$_K$ residues (F28, R31, E68, H70) make no direct contacts with rsCSP (Fig. 3b; Supplementary Table 2), indicating that the minor repeat region facilitates antibody-antibody affinity maturation in the context of multiple adjacent NPNV motifs, as has been observed for extended NPNA repeats[13,18,19].

L9 is one of the most potent anti-PfCSP mAbs identified to date and is currently undergoing clinical development as a monoclonal therapy for malaria prevention[4]. Thus, these structural data will be useful for rational antibody engineering to improve both the

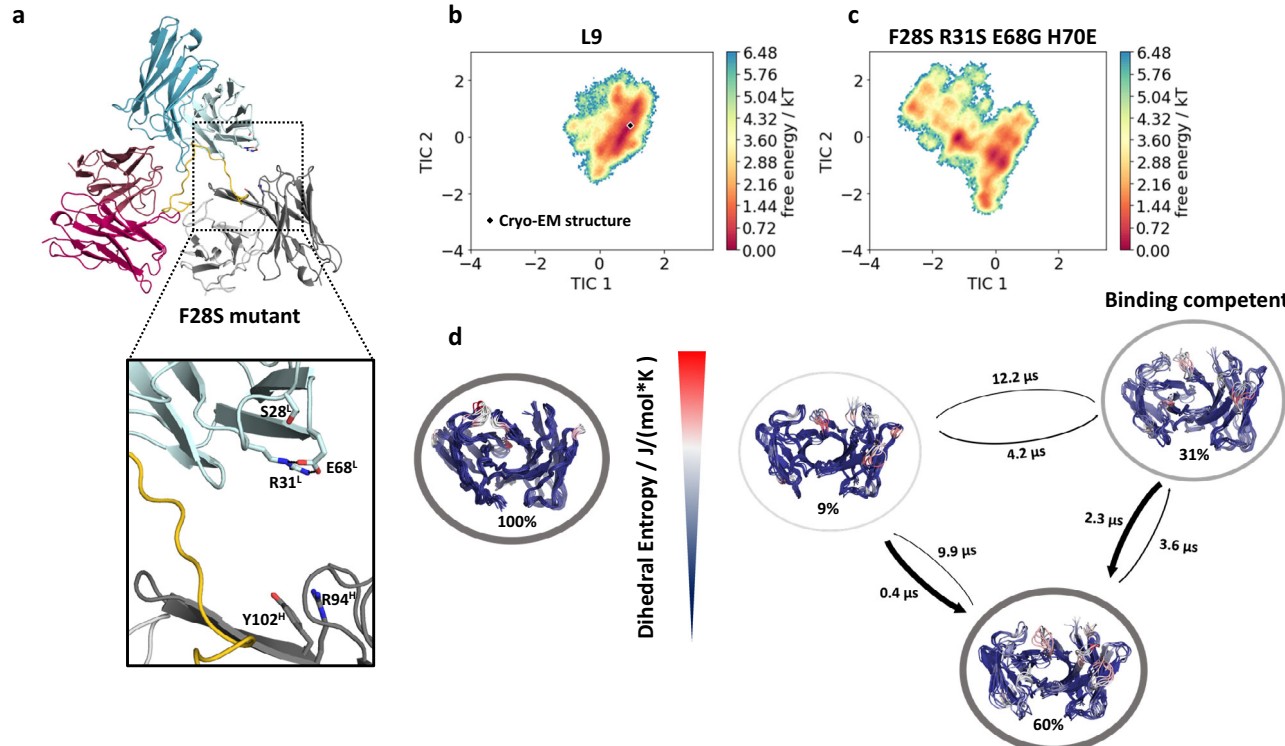

**Fig. 5 | Molecular dynamics reveals L9$_K$ residues critical for stability of the homotypic interface and PfCSP binding. a** Most populated structure for the F28S variant, highlighting the loss of critical homotypic interactions, which occurred in 74% of simulated structures. These contacts were maintained in 26% of F28S simulations. **b**, **c** Free energy landscapes of the L9 WT and the F28S/R31S/E68G/H70E variant projected in the same coordinate system, revealing a substantial increase in conformational space and a population shift due to the mutations. Cryo-EM structure is depicted as black diamond. $k$: Boltzmann constant; $T$: temperature. **d** Conformational ensemble representatives, state probabilities, and transition kinetics for the WT and the quadruple mutant, color-coded according to their dihedral entropy (blue-low flexibility, red-high variability). This mutant contains all four key homotypic contacts in the light chain mutated to the germline sequence (*IGKV1-5*). Note that H70 is E70 in fully germline *IGKV1-5* sequence, and D70 in the L9 precursor F10$_K$. *J*: joule; *K*: kelvin.

protective efficacy and pharmacokinetic properties of this mAb. The discovery of L9 and the NPNV minor repeat region as a highly protective epitope on PfCSP has led to new efforts to re-design PfCSP-based vaccines to elicit L9-like antibodies[22,23]. The cryo-EM structure presented here now enables a structure-based approach, which may be instrumental in developing the next-generation malaria vaccine. Future studies to identify related, NPNV-specific mAbs should enhance our understanding of this class of antibodies and their important contribution to protective immunity against malaria.

## Methods
### Protein production
L9 heavy and light chain sequences were synthesized and codon-optimized for mammalian expression and cloned into pHCMV3 by Genscript Inc. The full variable domain ($V_H/V_L$) and the light chain constant domain and heavy chain constant domain 1 ($C_L$ and $C_H1$) were included in each construct (heavy chain residues 1–216; light chain residues 1–214). The recombinant Fab was expressed by transient transfection in Freestyle 239F cells (ThermoFisher, cat #R79007) grown in Freestyle 239 Medium without antibiotics (Gibco, cat #12338018); during expression, Fab was secreted into the medium due to the N-terminal signal peptide. Cells were pelleted seven days after transfection. The medium was then filtered and run over a HiTrap KappaSelect (Cytivia; cat #17545812) affinity column followed by cation exchange chromatography (Mono S, Cytivia cat #11001287). All purification steps were performed in TBS (pH 8.0). rsCSP, which contained a C-terminal 6x-His tag, was cloned into the pET28a plasmid and expressed in the Shuffle strain of *E. coli* (New England Biolabs; cat #C3026J). The rsCSP construct contains residues 26–159 and 240–383

of the 3D7 strain PfCSP protein sequence (UniProt Q7K740), and is identical to the 3D7 sequence over these positions. rsCSP was purified as previously described[24]. Briefly, E. coli SHUFFLE competent cells were transformed with the rsCSP-pET28a plasmid, and a single colony was picked for a 50 mL overnight starter culture grown in LB broth supplemented with 50 ug/mL kanamycin. Two 1 L cultures were inoculated the next day with 25 mL each of the overnight culture, and were grown at 37 °C in LB supplemented with 50 ug/mL kanamycin. When the optical density at 600 nm reached a value of 1, the cultures were induced with 1 mM isopropyl β-ᴅ-1-thiogalactopyranoside (Sigma; cat #16758) for 6 h. The cells then were harvested and lysed by microfluidization in PBS (pH 7.4). The lysate was incubated overnight with Ni cOmplete resin (Sigma; cat # 5893682001) and was eluted in PBS (pH 7.4) containing 200 mM imidazole.

### Cryo-EM sample preparation
To form the L9 Fab-rsCSP complex, >10 fold molar excess of L9 Fab was incubated with rsCSP in tris-buffered saline (TBS; pH 8.0) overnight at 4 °C; this would theoretically allow for full Fab occupancy of the entire minor and major repeat regions on rsCSP. The complex was purified by size exclusion chromatography (SEC) with a Superdex 200 Increase 10/300 GL column (Sigma-Aldrich; cat #GE28-9909-44) equilibrated with TBS. All complex-containing fractions were pooled and used for structural studies. For initial cryo-EM attempts, the purified complex was concentrated to ~1 mg/mL with a 30 kDa molecular weight cutoff filter (MilliporeSigma; cat #MRCF0R030), and 3 μL of this solution was applied to holey gold UltrAufoil (Quantifoil) cryo-EM grids. Grids were blotted for two to four seconds at 100% humidity, 4 °C, and plunge-frozen with a Vitrobot Mark IV into liquid ethane. Due

to extensive aggregation of the complex during vitrification, and concomitant preferred orientation in vitrified ice, cryo-EM data were later collected with L9-rsCSP captured onto graphene oxide (GO) grids. For GO grid preparation, the L9-rsCSP complex was diluted to ~0.05 mg/mL in TBS, and 3 μL was applied to holey gold UltrAufoil grids containing a non-uniform layer of GO sheets on top of the grid. GO grids were made in-house; the fabrication and preparation procedure was adapted from a published protocol[25]. Briefly, UltrAufoil 1.2/1.3 holey gold grids (300 mesh) were washed with chloroform and allowed to dry completely. Grids were then glow-discharged, and 4 μL of 1 mg/mL PEI solution (polyethylenimine HCl, 25 mM HEPES pH 7.9) was applied to the grid and incubated for 2 min. Excess PEI was blotted with filter paper. Grids were washed with 2 drops of milli-Q water and allowed to dry completely. GO sheets (Sigma-Aldrich; cat #763705) were diluted to 0.2 mg/mL in water and centrifuged at $1500 \times g$. 4 μL of the supernatant was applied to grids and incubated for 2 min. Excess GO solution was blotted off, and grids were washed two times with water. Grids were allowed to dry for at least 30 min before use, and were used for sample vitrification on the same day they were prepared. This procedure resulted in ~90% coverage of the holes with GO; about half of these holes contained a monolayer of GO. For vitrification, 3 μL of L9-rsCSP complex (0.05 mg/mL) was applied to GO grids, and the sample was blotted for 2 s at 100% humidity, 4 °C, and plunge-frozen in liquid ethane.

### Cryo-EM data collection
Automated data acquisition was performed with the Leginon software[26] (version 3.5) on a Titan Krios (ThermoFisher) operated at 300 keV. Micrograph movies were collected in electron counting mode with a K2 Summit direct electron detector (Gatan), with an unbinned pixel size of 1.045 Å and a defocus range of −0.9 μm to −2.0 μm. The dose rate was ~6 e⁻/Å²/sec, with a full exposure time of 10 s; 200 ms per movie frame. This resulted in a total dose of ~60 e⁻/Å² on the specimen. A total of 12,521 movies were collected over four separate data sets. Due to preferred orientation of the L9-rsCSP complex on GO grids, two of these four data sets were collected with a stage tilt of −40°, with all other imaging parameters held constant. Movies and micrographs were cataloged and stored with the aid of Appion[27].

### Single particle cryo-EM data processing
Movie frames were aligned and dose-weighted with MotionCor2[28]. Subsequent processing was performed with cryoSPARCv3.3[29]. The contrast transfer function (CTF) was calculated with the Patch CTF Estimation tool, which was critical for accurate estimation of the tilted micrographs. The Gaussian (blob) picker was used on a subset of micrographs for initial particle picking, and 2D templates were generated with multiple rounds of 2D classification. Template picking was then used on the full dataset. Multiple rounds of 2D classification resulted in a particle stack containing 842,590 particles. A starting model generated from ab initio reconstruction was used for a non-uniform refinement job to achieve a resolution of ~3.7 Å. Multiple rounds of global CTF (beam tilt) refinement and per particle defocus refinement led to a 3.35 Å map. To account for possible flexibility between each of the three L9 Fabs, 3D Variability Analysis was used specifying four principal modes[30]. The output was fed into a 3D Variability Display job in cluster mode, specifying 20 clusters. Close inspection of the interactive cluster plots and structural comparison of cluster maps identified rotational flexibility in the Fab (Fab A) bound at the N-terminus of the peptide relative to the other two Fabs. The most homogeneous clusters were pooled, yielding a particle stack with 451,712 particles. These were again subjected to non-uniform and CTF refinement, leading to a 3.36 Å map with significantly improved interpretability of high-resolution features, particularly for the antigen (rsCSP) density.

To further improve the quality of the reconstruction of the rsCSP epitope and L9 Fab paratope structures, we masked the region surrounding the antigen for Local Refinement in CryoSPARC v4.0 using the following parameters: (1) a dilation radius of 5 and (2) a soft padding width of 10 (highlighted in orange in Supplemental Fig. 1g). All 842,509 particles that were used for the consensus refinement (Supplemental Fig. 1c), were used here. The Local refinement job generated a 3.34 Å map that enabled us to improve the overall density of the antigen and model interpretability, highlighted by the black circle in the second row.

### Atomic model building
The X-ray structure of 239 Fab bound to NPNA₂ (6W00), which contains matching germline heavy chain (*IGHV3-33*) and light chain (*IGKV1-5*) genes, was used to generate a homology model of L9 Fab. This model was then used as the template for re-building of the structure with RosettaCM[31]. At first, only the central Fab was modeled. On the resulting lowest energy model, the CDR loops were removed and built manually in Coot[32]. This structure was docked into the density of the two neighboring Fabs, and the trimeric Fab complex was refined with PHENIX real-space refine[33] (v1.20.1). Based on the known preferred epitope of L9, and inspection of the cryo-EM density, the structure of the PfCSP minor repeat region was built manually in Coot. The L9 Fab-rsCSP complex was again refined with PHENIX and errors were iteratively corrected with Coot. Rosetta Relax was used for a final all-atom refinement[34].

### Structural analysis
BSA and root mean square deviation (RMSD) calculations were performed in UCSF Chimera[35]. For general structural interpretation, UCSF Chimera and Coot were used. Calculation of electrostatic potential surfaces was performed with PyMol (The PyMOL Molecular Graphics System, Version 2.0 Schrödinger, LLC). The Epitope Analyzer webtool was used to assess direct contacts within the homotypic interface and between L9 Fab and rsCSP[36]. Structure figures were made with UCSF Chimera, UCSF ChimeraX[37], and PyMol.

### Molecular dynamics simulations
Based on the cryo-EM structure of the WT L9 (this study), containing three Fvs bound to rsCSP, we performed five replicas each of 1 μs of classical molecular dynamics simulations of the complex to identify critical residues that stabilize/favor the homotypic interface. For the other investigated variants (Supplementary Table 4), we derived the starting structures for our simulations from the WT L9 structure by replacing the respective amino acids, followed by a local energy minimization in MOE (Molecular Operating Environment, Chemical Computing Group, version 2020.09). The starting structures for simulations were prepared in MOE using the Protonate3D tool[38]. To neutralize the charges, we used the uniform background charge, which is required to compute long-range electrostatic interactions[39]. Using the tleap tool of the AmberTools20[40] package, the structures were soaked in cubic water boxes of TIP3P water molecules with a minimum wall distance of 12 Å to the protein[41,42]. For all simulations, parameters of the AMBER force field 14SB were used[43]. Molecular dynamics simulations were performed in an NpT ensemble using pmemd.cuda[44]. Bonds involving hydrogen atoms were restrained by applying the SHAKE algorithm[45], allowing a time step of 2 fs. Atmospheric pressure of the system was preserved by weak coupling to an external bath using the Berendsen algorithm[46]. The Langevin thermostat was used to maintain the temperature during simulations at 300 K. The interaction energies were calculated with cpptraj by using the linear interaction energy (LIE) tool[40]. We calculated the electrostatic and van der Waals interaction energies for all frames of each simulation (10000 frames/simulation) and provided the simulation-averages of these interaction energies in Supplementary Table 4.

A previously published method characterizing the CDR loop ensembles in solution[47] was used to investigate the conformational diversity of the six CDR loops of the free (apo) L9 Fv and the respective variants. To enhance the sampling of the conformational space, well-tempered bias-exchange metadynamics[48,49] simulations were performed in GROMACS[50,51] with the PLUMED 2 implementation[52]. We chose metadynamics as it enhances sampling on predefined collective variables (CV). The sampling is accelerated by a history-dependent bias potential, which is constructed in the space of the CVs[53]. As collective variables, we used a well-established protocol, boosting a linear combination of sine and cosine of the ψ torsion angles of all six CDR loops calculated with functions MATHEVAL and COMBINE implemented in PLUMED 2[47]. As discussed previously, the ψ torsion angle captures conformational transitions comprehensively[54]. The underlying method presented in this paper has been validated in various studies against a large number of experimental results[47,55]. The simulations were performed at 300 K in an NpT ensemble using the GPU implementation of the pmemd module[44] to be as close to the experimental conditions as possible and to obtain the correct density distributions of both protein and water. We used a Gaussian height of 10.0 kJ/mol and a width of 0.3 rad. Gaussian deposition occurred every 1000 steps and a biasfactor of 10 was used. 500 ns of bias-exchange metadynamics simulations were performed for the prepared Fv structures. The resulting trajectories were aligned to the whole Fv and clustered with cpptraj[40] using the average linkage hierarchical clustering algorithm with a RMSD cutoff criterion of 1.2 Å resulting in a large number of clusters. The cluster representatives for the antibody fragments were equilibrated and simulated for 100 ns using the AMBER 20 simulation package. The accumulated simulation times for the investigated L9 variants are summarized in Table S5.

With the obtained trajectories, we performed a time-lagged independent component analysis (tICA) using the python library PyEMMA 2 employing a lag time of 10 ns. tICA was applied to identify the slowest movements of the investigated Fv fragments and consequently to obtain a kinetic discretization of the sampled conformational space[56]. tICA is a dimensionality reduction technique that detects the slowest-relaxing degrees of freedom and facilitates kinetic clustering, which is a crucial pre-requisite for building a Markov-state model. It linearly transforms a set of high-dimensional input coordinates to a set of output coordinates, by finding a subspace of *"good reaction coordinates"*. Thereby, tICA finds coordinates of maximal autocorrelation at a given lag time. The lag time sets a lower limit to the timescales considered in the tICA and the Markov-state model. Accordingly, tIC1 and tIC2 represent the two slowest degrees of freedom of the systems.

Based on the tICA conformational spaces, thermodynamics and kinetics were calculated with a Markov-state model (MSM)[57] by using PyEMMA 2, which uses the k-means clustering algorithm to define microstates and the PCCA+ clustering algorithm[58] to coarse-grain the microstates to macrostates. Markov-state models are network models which provide valuable insights for conformational states and transition probabilities between them, as it allows identification of the boundaries between two states[57]. Basically, MSMs coarse-grain the system's dynamics, which reflect the free energy surface and ultimately determine the system's structure and dynamics. Thus, MSMs provide important insights and enhance the understanding of states and transition probabilities and facilitates a quantitative connection with experimental data[59].

The sampling efficiency and the reliability of the Markov-state model (e.g., defining optimal feature mappings) has been evaluated with the Chapman-Kolmogorov test by using the variational approach for Markov processes and monitoring the fraction of states used, since the network states must be fully connected to calculate probabilities of transitions and the relative equilibrium probabilities. To build the Markov-state model, we used the backbone torsions of the respective CDR loops, defined 100 microstates using the k-means clustering algorithm and applied a lag time of 10 ns.

Additionally, we calculated the residue-wise dihedral entropies with the recently published X-entropy python package, which calculates the entropy of a given dihedral angle distribution[60]. This approach uses a Gaussian kernel density estimation (KDE) with a plug-in bandwidth selection, which is fully implemented in C++ and parallelized with OpenMP. The obtained residue-wise dihedral entropies were projected onto the respective structures (Fig. 5d).

**Biolayer interferometry (BLI)**
To evaluate the binding of L9 to the major and minor PfCSP repeats, BLI experiments were performed using the Octet Red96 system (ForteBio). A basic kinetics experiment was used to measure interaction of L9 and 311 Fabs to $NPNA_8$ (major repeat only) and rsCSP (major + minor repeats). 311 was used as a positive control for $NPNA_8$ binding. Kinetics buffer (PBS + 0.01% BSA, 0.002% Tween-20, pH 7.4) was used for all dilutions, baseline measurements, and reference subtractions. Biotinylated $NPNA_8$ or Twin-Strep tagged rsCSP was diluted to 5 μg/mL in kinetics buffer (KB) and immobilized onto Streptavidin BLI biosensors (Sartorius). Binding kinetics for each antibody were measured across a dilution series comprising the following concentrations of Fab (in nM): 6.25, 12.5, 25, 50, 100, 200. The steps of the kinetics experiment were as follows: baseline, 60 s (KB only), antigen loading, 600 s (KB + antigen), baseline 2, 60 s (KB only), association, 600 s (KB + antibody), dissociation, 1200 s (KB only). BLI data were processed with the ForteBio Data Analysis 9.0 software to evaluate kinetic parameters. In each case, global (full) fitting was performed with a 2:1 binding model, as there were at least two binding sites per peptide that are likely non-independent (4 sites for $NPNA_8$, 11 sites for rsCSP), and a 1:1 kinetic model yielded substantially lower $R^2$ values. Using a 2:1 kinetic model, two $K_D$ values are reported; for comparison across mAbs and peptides, an overall affinity to each peptide was calculated as an average of these two values, which were in turn averaged across at least 4 concentrations of Fab with an $R^2$ of ≥0.95.

**Reporting summary**
Further information on research design is available in the Nature Portfolio Reporting Summary linked to this article.

## Data availability
The coordinates for the L9-rsCSP structure and corresponding cryo-EM map generated in this study have been deposited in the Protein Data Bank (PDB) and Electron Microscopy Data Bank (EMDB) under accession codes 8EH5 and EMD-28135, respectively. All other antibody structures used for comparison with L9 in Fig. 2e, Fig. 3e–g, and Supplementary Figs. 3 and 5, were obtained from previous studies and deposited to the PDB under the following accession codes: 6D01 (1210-NANP5), 6O23 (2243-NANP5), 6O24 (4498-NANP3), 6ULE (2541-NANP5), 6W00 (239-NPNA2), 6WFW (364-NPNA2), 7RQQ (F10H/L9k-NPNV) and 7RQR (L9H/F10k-NPNV). Source data are provided with this paper.

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

## Acknowledgements

The authors thank B. Anderson for maintenance and administration of the cryo-EM facility at The Scripps Research Institute, and H.L. Turner and C.A. Bowman for technical support. We also thank L.T. Wang and N.K. Hurlburt for sharing of reagents and insightful discussions, and J.R. Riccabona and Y. Wang for fruitful discussions and technical support. The computational results presented here have been achieved (in part) using the Vienna Scientific Cluster (VSC). We acknowledge PRACE for awarding us access to Piz Daint at CSCS, Switzerland. The research has been supported by the National Institutes of Health grant 1F32AI150216-01A1 (GMM), The Bill and Melinda Gates Foundation grant INV-004923 (IAW, ABW), the Austrian Academy of Sciences APART-MINT postdoctoral fellowship and the Austrian Science Fund grant: P34518 (MFQ).

## Author contributions

G.M.M., M.P., R.A.S., I.A.W., and A.B.W. conceived the project. G.M.M., M.F.Q,. W.H.L., and T.P. designed and performed experiments, and analyzed the data. L.E.W. analyzed data. K.R.L., M.P., R.A.S., I.A.W., and A.B.W. acquired funding and supervised the project. G.M.M. and M.F.Q. wrote the original manuscript draft. All authors contributed to the manuscript review and editing.

## Competing interests

The authors declare no competing interests.
