## [Peer Review File · Nature Communications]

Structural basis of epitope selectivity and potent protection from malaria by PfCSP antibody L9REVIEWER COMMENTS

Reviewer #1 (Remarks to the Author):

In the current report, Martin et al. describe the cryo-EM structure of three L9 Fab fragments bound to the minor (NPNV) repeats of a recombinant, shortened construct of PfCSP, rsCSP. The authors uncover the molecular basis of monoclonal antibody (mAb) L9's selectivity and high affinity for NPNV repeats and delineate homotypic interactions between adjacent Fabs binding to consecutive NPNV motifs interspersed with DPNA sequences. Their structural description provides additional insight into the previously observed critical role of the L9 kappa light chain (L9k) for PfCSP recognition, as the light chain contributes key interactions and over half of the Fab BSA involved in both rsCSP binding and the homotypic interface. Importantly, the paratope structure of L9 is distinct from most other IGHV3-33 mAbs targeting PfCSP major and minor repeats. By conducting molecular dynamics simulations on L9 Fab variants in which up to four specific L9k residues that engaged in homotypic interactions but did not contact rsCSP were germline-reverted or mutated to a predicted precursor amino acid, the authors reveal that these residues play a key role in maintaining L9 Fab-rsCSP immune complex stability.

Altogether, this investigation builds on previous studies characterizing mAb L9 and provides critical insights into the molecular binding properties of this potent mAb. In a recent phase I clinical trial, mAb L9 emerged as a promising antibody therapeutic candidate for malaria prevention. Thus, in addition to guiding future antibody engineering efforts and next-generation immunogen design, the findings presented here further support the clinical development of mAb L9. Therefore, due to the significant implications of this study, this reviewer recommends this manuscript for publication, provided that the authors address the following concerns.

Major points:

- 1) The authors indicate that the L9 Fab-rsCSP complex was generated by incubating an excess of L9 Fab with rsCSP and purified through size exclusion chromatography on a Superdex200 Increase column. Purification on this column may exclude any co-complexes with more than three Fabs bound to one rsCSP molecule based on high molecular weight (i.e., these may elute in the void volume). Because mAb L9 was previously shown to bind a construct of PfCSP lacking all NPNV repeats, albeit with slightly lower affinity (Wang et al. *Immunity* 2020), this Reviewer is left wondering whether an L9 Fab-rsCSP complex with more than three Fabs bound is possible (i.e., with Fabs bound to major repeat motifs as well as minor repeats). To substantiate their focus on the structure of a "trimeric Fab-rsCSP complex", it would be important to 1) perform ITC and/or SEC-MALS with a large stoichiometric excess of L9 Fab relative to all minor and major repeats and indeed prove that no more than three Fabs bind to rsCSP – i.e., the NPNA repeats remain unoccupied; and 2) mention whether 2D-classes were observed with more than three Fabs bound to rsCSP and if so, document their frequency and discuss the implications of this observation.
- 2) It is stated that: "With this unique CDR conformation, L9 appears optimally disposed to bind the bulkier minor repeat residue V4, which is the only difference between the NPNA and NPNV epitopes." This needs to be expanded: what is the molecular basis that prevents L9 from binding NPNA repeats – i.e., make L9 specific for NPNV?
- 3) It would also be beneficial to the reader for a map to be presented in a Supp Fig. for rsCSP residues modeled, particularly for those residues that would be different if a different register was used – i.e., show the confidence (experimental evidence from the map) in assigning NPNV instead of DPNA or NPNA. Indeed, the authors state: "This basic binding pocket is nearly fully occupied by the N3 side chain, which forms key H-bonds with R96L". Is it possible that an alternate register modeled puts a PfCSP Asp side chain in this position instead making salt bridge interactions with R96L, which energetically might be more favorable?
- 4) In a previous report, F10 was identified as a clonal relative and predicted precursor to L9. In the current study, the F10 kappa light chain (F10k) inspired the generation of several L9 Fab variants that are used to investigate the effects of somatic hypermutation in the L9 light chain. Thus, to clarify the rationale of these variants to a more general reader, it would be beneficial for the authors to describe the relevance of F10 in more detail.

Minor points:

- 1) In the Abstract, the authors mention: "novel set of affinity-matured homotypic". It is unclear

"novel" relative to what? Is this qualifier necessary?

- 2) The authors note that the H70D substitution stabilizes the homotypic interface based on predicted interaction energies (line 168); however, based on the data presented in Fig. S6, the H70D variant appears to sample a broader energy landscape than the L9 WT. Can the authors comment on this discrepancy?
- 3) Line 44-46: a citation for this sentence would strengthen the author's statement regarding the benefits of potent mAbs.
- 4) Line 53: "CIS43LS"
- 5) Line 75-76: the authors should clarify whether the RMSD values indicated here are all-atom or backbone measurements and include a description of these RMSD values in the Fig. S2 legend as well.
- 6) Line 82: and light chain lineages?
- 7) Line 85-86: an additional reference to Fig. S3A-B here would be helpful to indicate the relative contributions of the CDRs to PfCSP binding.
- 8) Line 87-88: "550 Å²" and "256 Å²"
- 9) Line 104: a reference to Fig. 2E here would be appropriate, given the comparison to the other IGHV3-33-encoded mAbs.
- 10) Line 126: a reference to Fig. 3B-D may be more appropriate here.
- 11) Line 147: "The four somatic mutations in L9K are atypical: F28L, E68L, and H70L": it is unclear why there is both K and L nomenclature for the light chain.
- 12) Fig. 4A and Movie S1: to strengthen their message, it may be helpful for the authors to indicate the frequency at which the F28S mutant adopts structures maintaining key hydrogen bond and salt bridge interactions, compared to that for structures in which these interactions are broken.
- 13) Fig. 4B-C: it is unclear why these plots are not identical to the corresponding diagrams in Fig. S6. In the Fig. 4 legend and Fig. S6, there also appears to be a typo: H70E should be H70D.
- 14) Fig. S1C legend: "Left: representative cluster maps from 3D variability analysis"
- 15) Fig. S3A legend: "Sequence alignment to the IGHV3-33 germline gene"
- 16) Fig. S3E and F: it is unclear why the structure of mAb 2243 is in a separate figure panel.
- 17) Movie S1: it would be beneficial to include a stick representation of F100cH, as this residue forms a cation-pi interaction with R31L in the L9 WT homotypic interface.
- 18) Materials and Methods: MonoS contains a typo.
- 19) Use PfCSP consistently as an abbreviation throughout the manuscript; sometimes it is only CSP: e.g., line 152, 204 nM vs 13 nM for CSP peptides, in Figure legends, etc.

Reviewer #2 (Remarks to the Author):

This study describes the cryo-EM structure of a therapeutically important antibody, L9, in complex with its PfCSP repeat epitope. The manuscript is well-written, and the analysis and figures appropriately describe the key features of the complex, including homotypic interactions and comparisons with other structures. This study and the structure will likely be of high interest to the community, adding very useful information to the malaria immunotherapy and vaccine field. There are only minor comments to address:

1. Lines 75-76: "Relative to the L9 cryo-EM structure, RMSD values for both chimeric Fvs are ~0.5Å, and ~0.1Å over the NPNV peptide." This statement provides useful information, but as stated it is somewhat unclear regarding the comparisons and can be re-worded (the supplemental figure is more clear regarding this, but ideally readers should not need to refer to that). The cryo-EM structure contains three structures of L9, thus the authors can re-word to note that a representative Fab/Fv was used. Additionally, the atoms being used in the RMSD (Ca or backbone) do not seem to be noted in the main text or methods, and ideally that should be stated in both.
2. Related to the above comment, it does not seem that RMSDs between the different Fv's in the cryo-EM structure are given. The RMSDs are likely small, but it is possible that there is still some heterogeneity among the L9 copies in the structure. If those RMSDs have not been given, they should be noted, along with the epitope RMSDs.

3. Line 168: "Interestingly, the H70D single mutant stabilizes the homotypic interface (table S4)."
This statement seems to note the H70D stabilization as fact, whereas it is the result of computational simulations. While it is apparent from context that computational simulations are being performed, it would be best to avoid ambiguity by re-wording this sentence, e.g. "Interestingly, the H70D single mutant is predicted to stabilize...". Other sentences describing the computational predictions in this paragraph can potentially also be re-worded for clarity on this point. Relatedly, the title of Table S4, "Interaction energies of homotypic interface..." can be modified to reflect that these are computational predictions, e.g. "Computed interaction energies...", "Predicted interaction energies...", or "MD simulation interaction energies...".

REVIEWER COMMENTS

Reviewer #1 (Remarks to the Author):

In the current report, Martin et al. describe the cryo-EM structure of three L9 Fab fragments bound to the minor (NPNV) repeats of a recombinant, shortened construct of PfCSP, rsCSP. The authors uncover the molecular basis of monoclonal antibody (mAb) L9's selectivity and high affinity for NPNV repeats and delineate homotypic interactions between adjacent Fabs binding to consecutive NPNV motifs interspersed with DPNA sequences. Their structural description provides additional insight into the previously observed critical role of the L9 kappa light chain (L9k) for PfCSP recognition, as the light chain contributes key interactions and over half of the Fab BSA involved in both rsCSP binding and the homotypic interface. Importantly, the paratope structure of L9 is distinct from most other IGHV3-33 mAbs targeting PfCSP major and minor repeats. By conducting molecular dynamics simulations on L9 Fab variants in which up to four specific L9k residues that engaged in homotypic interactions but did not contact rsCSP were germline-reverted or mutated to a predicted precursor amino acid, the authors reveal that these residues play a key role in maintaining L9 Fab-rsCSP immune complex stability.

Altogether, this investigation builds on previous studies characterizing mAb L9 and provides critical insights into the molecular binding properties of this potent mAb. In a recent phase I clinical trial, mAb L9 emerged as a promising antibody therapeutic candidate for malaria prevention. Thus, in addition to guiding future antibody engineering efforts and next-generation immunogen design, the findings presented here further support the clinical development of mAb L9. Therefore, due to the significant implications of this study, this reviewer recommends this manuscript for publication, provided that the authors address the following concerns.

Major points:

1) The authors indicate that the L9 Fab-rsCSP complex was generated by incubating an excess of L9 Fab with rsCSP and purified through size exclusion chromatography on a Superdex200 Increase column. Purification on this column may exclude any co-complexes with more than three Fabs bound to one rsCSP molecule based on high molecular weight (i.e., these may elute in the void volume). Because mAb L9 was previously shown to bind a construct of PfCSP lacking all NPNV repeats, albeit with slightly lower affinity (Wang et al. *Immunity* 2020), this Reviewer is left wondering whether an L9 Fab-rsCSP complex with more than three Fabs bound is possible (i.e., with Fabs bound to major repeat motifs as well as minor repeats). To substantiate their focus on the structure of a "trimeric Fab-rsCSP complex", it would be important to 1) perform ITC and/or SEC-MALS with a large stoichiometric excess of L9 Fab relative to all minor and major repeats and indeed prove that no more than three Fabs bind to rsCSP – i.e., the NPNA repeats remain unoccupied; and 2) mention whether 2D-classes were observed with more than three Fabs bound to rsCSP and if so, document their frequency and discuss the implications of this observation.

The Reviewer is correct that Wang et al (*Immunity* 2020) showed that L9 can bind both the minor and major repeats, with affinities of 1.9 nM and 73 nM (by ITC with IgG), respectively. Thus, it is conceivable that complexes of L9 Fab could show both major and minor repeat occupancy. However, we are confident that the trimeric L9 Fab-rsCSP complex we identified is the primary species, and that no larger complexes with NPNA binding were present in our 3D classes.

1. In forming the complex for SEC and cryo-EM, we used a large molar excess of L9 Fab relative to rsCSP (>10 fold), which is full occupancy for this antigen (rsCSP contains 11 total minor and major repeat

epitopes consisting of either NPNANPNA or DPNANPNV). *We have now specified this in the Methods (lines 279-281).*

2. Based on the manufacturer's specifications (Cytiva) a Superdex200 increase column has a fractionation range of approximately 5 to 600 kDa. The trimeric L9 Fab-rsCSP complex we identified is ~180kDa (50 kDa * 3 Fabs + 30 kDa rsCSP). Thus, this is a sufficient range to capture complexes with more than three Fabs, which we did not observe. Further, in preparing the cryo-EM sample, all fractions containing complex were pooled, so it is unlikely we excluded higher MW species. *We have now specified this in the Methods (lines 282-284).*

3. In both NS-EM and cryo-EM, we did not observe any classes containing more than 3 Fabs. *We have now specified this in the Results (lines XX).*

4. To support our conclusions above, we used BLI to compare binding of L9 Fab to rsCSP and a peptide containing only NPNA repeats, NPNA₈. We see modest binding to NPNA₈ but rapid dissociation, distinct from the strong binding profile of L9 to rsCSP. Thus, the interaction of L9 with NPNA is transient, and unlikely to be maintained after SEC purification and therefore observable in our cryo-EM experiment. *We have included a statement summarizing these data in the Results (Lines 93-98).*

5. To summarize these conclusions, we have made a new supplementary figure, Supp. Fig. 2, which shows the SEC profile, BLI data, and a more exhaustive panel of 2D classes.

2) It is stated that: "With this unique CDR conformation, L9 appears optimally disposed to bind the bulkier minor repeat residue V4, which is the only difference between the NPNA and NPNV epitopes." This needs to be expanded: what is the molecular basis that prevents L9 from binding NPNA repeats – i.e., make L9 specific for NPNV?

We agree with the reviewer that this is a key point. As such, we have expanded on this section in the Results to provide a more detailed explanation of L9 npnv/npna selectivity based on our structural data (Lines 139-148). We have also made an additional supplementary figure (Supp. Fig. 5) to support our conclusions.

3) It would also be beneficial to the reader for a map to be presented in a Supp Fig. for rsCSP residues modeled, particularly for those residues that would be different if a different register was used – i.e., show the confidence (experimental evidence from the map) in assigning NPNV instead of DPNA or NPNA. Indeed, the authors state: "This basic binding pocket is nearly fully occupied by the N3 side chain, which forms key H-bonds with R96L". Is it possible that an alternate register modeled puts a PfCSP Asp side chain in this position instead making salt bridge interactions with R96L, which energetically might be more favorable?

To support our model of CSP in the cryo-EM structure, we have now made a new supplementary figure, Supp. Fig. 4, which shows the fit to the density of our original model and test models with altered registration, such that either DPNA or NVDP is the core motif within the L9 binding pocket (as opposed to NPNV). We have also included a statement in the Results summarizing the key arguments in favor of our model of CSP, which is centered on NPNV (lines 109-113). Overall, the two alternate registrations suggested by the reviewer above are highly unlikely, for the following reasons:

1. As has been shown in numerous publications, the repeating structural unit in both the minor and major repeats is DPNA, NPNA, or NPNV (Oyen et al. Sci Advances 2018; Pholcharee et al. Nat Comms 2021; Kosalu et al. Nat Med 2018; Wang et al. Cell Rep 2022); each of these repeats form type 1 beta

turns as the proline-mediated kink allows H-bonding between the Asp1/Asn1 side chain and the main chain of Asn3. Thus, the core motif interacting with L9 is likely to be NPNV or DPNA, with our previous publication (Wang, Hurlburt et al, Cell Reports 2022) demonstrating a strong selectivity of L9 for NPNV with both structural and functional experiments. To maintain the cadence of the type-1 beta turn, and to keep the Pro residues in the correct position (which produce obvious kinks in the cryo-EM density), the only other likely registration of CSP in our structure would place DPNA as the core interaction motif, and NPNV as the linker motif (which has very little interaction with L9). In both cases, it is N3, either from ¹DPNA⁴ or ¹NPNV⁴ which will interact with R96^L of L9. To produce the salt bridge between D1 and R96, which as the Reviewer suggests may in theory be more favorable than N3, the core motif in the L9 paratope becomes NVDP, and the linker NANP. This not only disrupts the type-1 beta turn and produces an epitope structure with a partial beta turn from both N-terminal and C-terminal repeats, which is unlikely, but also places each of the Pro residues in a new position, and not at the kinked locations within the density map. Therefore, this registration, centered on NVDP, is highly unlikely. The only other registration likely is thus centered on DPNA, which we consider below.

2. In both alternate registrations, the fit to the cryo-EM density is substantially worse, particularly for the model centered on NVDP.

3. The structure of the NPNV repeats in the L9 cryo-EM structure presented here are nearly identical to the NPNV repeat structure in the X-ray structures of the L9 chimeric mAbs, in which either the heavy chain or light chain of L9 was replaced with the corresponding chain from F10, a recent precursor mAb of L9 isolated from the same individual (F10-Heavy/L9-Light and L9-Heavy/F10-Light; Wang, Hurlburt et al, Cell Reports 2022). The RMSD values between the NPNVs in the cryo-EM structure and the NPNV peptide in the L9 chimera X-ray structures are 0.05 and 0.1Å. This is shown in Figure S2, and confirms the validity of our structure, which was built solely on our cryo-EM density.

4. Based on the L9 cryo-EM structure, the DPNA repeat acts as a linker between NPNV repeats, and contributes almost no direct interactions with L9 Fab (Fig 1G). This suggests that this intervening sequence, either DPNA as modelled, or NPNV with the alternate registration, would have very low/no binding to L9. Thus, given that the authors were able to obtain crystal structures of these L9 chimeras in complex with a peptide of the sequence NANPNVDP, suggests that the interaction of L9 with NPNV is relatively high affinity, and is likely the core interaction motif within the minor repeat region.

5. In the above reference, the authors also demonstrate, by competition ELISA, that L9 is highly selective for NPNV, i.e. mutation of any residue within either of the two NPNV repeats contained within peptide 22 (NANPNVDPNANPNVD) caused significant loss of binding of L9 to peptide 22, whereas mutation of the DPNA sequence had almost no effect. Thus it seems improbable that DPNA would be the core motif bound to L9, as would be the case with the alternate registration.

4) In a previous report, F10 was identified as a clonal relative and predicted precursor to L9. In the current study, the F10 kappa light chain (F10k) inspired the generation of several L9 Fab variants that are used to investigate the effects of somatic hypermutation in the L9 light chain. Thus, to clarify the rationale of these variants to a more general reader, it would be beneficial for the authors to describe the relevance of F10 in more detail.

We thank the Reviewer for highlighting this key point. We have expanded on this section in the Results to provide greater context to the relevance of F10 (lines 191-200).

Minor points:

1) In the Abstract, the authors mention: “novel set of affinity-matured homotypic”. It is unclear “novel” relative to what? Is this qualifier necessary?

To avoid confusion, we have changed “novel” to “unique”

2) The authors note that the H70D substitution stabilizes the homotypic interface based on predicted interaction energies (line 168); however, based on the data presented in Fig. S6, the H70D variant appears to sample a broader energy landscape than the L9 WT. Can the authors comment on this discrepancy?

The H70D substitution stabilizes the homotypic interface in the context of the trimeric Fab-rsCSP complex. However, in simulations of the free (apo) Fv, this mutation results in a broader conformational space of the paratope. We have clarified this in the Results (Lines 215-220).

3) Line 44-46: a citation for this sentence would strengthen the author's statement regarding the benefits of potent mAbs.

We have added references demonstrating sterile protection from malaria in clinical trials of mAbs cis43LS (Gaudinski et al. NEJM 2021; Lyke et al. Lancet Inf Dis 2023) and L9LS (Wu et al. NEJM 2022)

4) Line 53: "CIS43LS"

Fixed

5) Line 75-76: the authors should clarify whether the RMSD values indicated here are all-atom or backbone measurements and include a description of these RMSD values in the Fig. S2 legend as well.

We have specified that reported RMSD values are for C α in both the Results and Figure S2.

6) Line 82: and light chain lineages?

Changed to "variety of heavy and light chain lineages"

7) Line 85-86: an additional reference to Fig. S3A-B here would be helpful to indicate the relative contributions of the CDRs to PfCSP binding.

Reference added here for Figure S4A and B (originally Fig. S3).

8) Line 87-88: "550 Å²" and "256 Å²"

"550Å²" and "256Å²" are correct

9) Line 104: a reference to Fig. 2E here would be appropriate, given the comparison to the other IGHV3-33-encoded mAbs.

Added reference to Fig. 2e

10) Line 126: a reference to Fig. 3B-D may be more appropriate here.

Fixed reference

11) Line 147: "The four somatic mutations in L9K are atypical: F28L, E68L, and H70L": it is unclear why there is both K and L nomenclature for the light chain.

We have provided a brief explanation of the nomenclature in the legend for Figure 2, which is as follows: "Note that in the main text and in figures, we use the nomenclature L9_K and L9_H to refer to the light chain and heavy chain, respectively, as a whole. When referring to specific amino acids within either chain, we use the more general notation of, for example, W32^L and W52^H."

12) Fig. 4A and Movie S1: to strengthen their message, it may be helpful for the authors to indicate the frequency at which the F28S mutant adopts structures maintaining key hydrogen bond and salt bridge interactions, compared to that for structures in which these interactions are broken.

We have added these values to legend of Figure 4a, which is as follows: “Most populated structure for the F28S variant, highlighting the loss of critical homotypic interactions, which occurred in 74% of simulated structures. These contacts were maintained in 26% of F28S simulations.”

13) Fig. 4B-C: it is unclear why these plots are not identical to the corresponding diagrams in Fig. S6. In the Fig. 4 legend and Fig. S6, there also appears to be a typo: H70E should be H70D.

We thank the reviewer for pointing this out. The plots are not identical, because in Fig S6 (now Supp. Fig. 8), all structures are projected in the same coordinate system for comparison. Figure 5B-C shows only WT and the mutant in the same coordinate system, which is required to properly calculate and visualize state probabilities and representatives. In this mutant, all positions are mutated to the germline IGKV1-5 gene, to understand how affinity maturation may have contributed to the development of homotypic Fab binding. Thus in this mutant H70 is Glu. In later simulations we use the H70D mutation, as seen in F10_k, to gauge differences between L9 and F10. We specified this in the Figure 5 legend.

14) Fig. S1C legend: “Left: representative cluster maps from 3D variability analysis”

Fixed

15) Fig. S3A legend: “Sequence alignment to the IGHV3-33 germline gene”

Fixed

16) Fig. S3E and F: it is unclear why the structure of mAb 2243 is in a separate figure panel.

We have added a statement in the legend of Fig. S3E (now Fig. 3) describing the main point of the figure, as follows: “NPNA₂ epitope structure in the NPNA-specific mAb 2243 (PBD 6O23), highlighting the two key CH- π interactions of germline encoded aromatic residues (W52^H and Y94^L) with the repeat prolines.”

17) Movie S1: it would be beneficial to include a stick representation of F100cH, as this residue forms a cation- π interaction with R31L in the L9 WT homotypic interface.

We have added a stick representation of residue F100c^H in Movie S1.

18) Materials and Methods: MonoS contains a typo.

Fixed

19) Use PfCSP consistently as an abbreviation throughout the manuscript; sometimes it is only CSP: e.g., line 152, 204 nM vs 13 nM for CSP peptides, in Figure legends, etc.

Fixed

Reviewer #2 (Remarks to the Author):

This study describes the cryo-EM structure of a therapeutically important antibody, L9, in complex with its PfCSP repeat epitope. The manuscript is well-written, and the analysis and figures appropriately describe the key features of the complex, including homotypic interactions and comparisons with other structures. This study and the structure will likely be of high interest to the community, adding very useful information to the malaria immunotherapy and vaccine field. There are only minor comments to address:

1. Lines 75-76: “Relative to the L9 cryo-EM structure, RMSD values for both chimeric Fvs are $\sim 0.5\text{\AA}$, and $\sim 0.1\text{\AA}$ over the NPNV peptide.” This statement provides useful information, but as stated it is somewhat unclear regarding the comparisons and can be re-worded (the supplemental figure is more clear regarding this, but ideally readers should not need to refer to that). The cryo-EM structure contains three structures of L9, thus the authors can re-word to note that a representative Fab/Fv was used. Additionally, the atoms being used in the RMSD (Ca or backbone) do not seem to be noted in the main text or methods, and ideally that should be stated in both.

We have re-worded this statement in the Results for clarity, specifying that comparisons with the F10 chimeras are relative to Fab B. (lines 100-103). We have also specified in the text and legend for Supp. Fig. 3 that RMSD values are for $C\alpha$.

2. Related to the above comment, it does not seem that RMSDs between the different Fv’s in the cryo-EM structure are given. The RMSDs are likely small, but it is possible that there is still some heterogeneity among the L9 copies in the structure. If those RMSDs have not been given, they should be noted, along with the epitope RMSDs.

We have now stated in the Results the $C\alpha$ RMSD values for the three L9 Fvs and the three NPNV epitopes in the cryo-EM structure (lines 103-105). We have also added additional panels to Supp. Fig 3 (Fig. S3e-g) to support these statements.

3. Line 168: “Interestingly, the H70D single mutant stabilizes the homotypic interface (table S4).” This statement seems to note the H70D stabilization as fact, whereas it is the result of computational simulations. While it is apparent from context that computational simulations are being performed, it would be best to avoid ambiguity by re-wording this sentence, e.g. “Interestingly, the H70D single mutant is predicted to stabilize...”. Other sentences describing the computational predictions in this paragraph can potentially also be re-worded for clarity on this point. Relatedly, the title of Table S4, “Interaction energies of homotypic interface...” can be modified to reflect that these are computational predictions, e.g. “Computed interaction energies...”, “Predicted interaction energies...”, or “MD simulation interaction energies...”.

We thank the Reviewer for pointing this out. We have reworded multiple statements throughout this paragraph (lines 205-243) to make clear that these are MD simulations, and are thus predictions. We have also reworded the title of Supplemental Table 4 to “Computed interaction energies of homotypic interface in trimeric L9-rsCSP complexes.”

REVIEWERS' COMMENTS

Reviewer #1 (Remarks to the Author):

The authors are to be congratulated for addressing all concerns raised during initial review with insightful new experimental insights.

My only remaining suggestion would be for the authors to consider revising the new language introduced in lines 144-149 to something that more closely matches the language described in Supp Fig. 5: "Structure of L9 paratope disfavors NPNA2 binding." e.g. "[...] the unique arrangement of the L9 CDR loops is unfavourable to NPNA binding in this conformation [...]".

REVIEWERS' COMMENTS

Reviewer #1 (Remarks to the Author):

The authors are to be congratulated for addressing all concerns raised during initial review with insightful new experimental insights.

My only remaining suggestion would be for the authors to consider revising the new language introduced in lines 144-149 to something that more closely matches the language described in Supp Fig. 5: "Structure of L9 paratope disfavors NPNA2 binding." e.g. "[...] the unique arrangement of the L9 CDR loops is unfavourable to NPNA binding in this conformation [...]".

We thank the Reviewer for this helpful suggestion, and have modified the statement in lines 144-149 as suggested. The new lines are as follows (now lines 133-138):

“Comparison of the paratope structure of L9 to a panel of NPNA-targeting *IGHV3-33/IGKV1-5* mAb structures suggests that, in addition, the unique arrangement of the L9 CDR loops is unfavorable to NPNA binding in this conformation, as superimposition of these Fab-NPNA₂ cryo-EM and X-ray structures onto the L9 Fab structure revealed extensive clashing between the peptide and the L9 CDRH1, CDRH3, and CDRL3 loops (Supplementary Fig. 5b, c).”